# Mapping of Ecological Corridors as Connections between Protected Areas: A Study Concerning Sardinia, Italy

Federica Isola [ID], Federica Leone *[ID] and Corrado Zoppi [ID]

Department of Civil and Environmental Engineering and Architecture, University of Cagliari, Via Marengo 2, 09123 Cagliari, Italy; federica.isola@unica.it (F.I.); zoppi@unica.it (C.Z.)
* Correspondence: federicaleone@unica.it; Tel.: +39-070-6755213

**Abstract:** According to the European Commission, green infrastructure and spatial connectivity concerning the provision of ecosystem services are strictly related to the conceptual category of ecological networks. In particular, regional and urban planning processes should adequately manage, improve and monitor the effectiveness of green infrastructures as ecological networks which provide ecosystem services and the spatial connectivity of such systems. Building on a methodological approach defined in previous studies, this article aims at identifying ecological corridors through a least-cost path model with reference to the spatial layout of a set of protected areas. Moreover, such a methodological approach is implemented in the context of the Sardinian region to map ecological corridors, which form, together with protected areas, a network representing the spatial framework of regional green infrastructure. Finally, the study discusses the relation between ecological corridors and the spatial taxonomy of the landscape components featured by environmental relevance, identified by the Regional Landscape Plan, through multiple linear regression analysis, in order to assess if, and to what extent, the present regional spatial zoning code can be used as a basis to implement regulations aimed at protecting ecological corridors. This methodological approach is relevant to defining planning policies and measures to strengthen the operational capacity and effectiveness of regional networks of protected areas through the protection and the improvement of the spatial framework of ecological corridors.

**Keywords:** ecological corridors; protected areas; landscape components; least-cost path model; multiple linear regression analysis



## 1. Introduction

This study aims at defining and implementing a methodology to identify ecological corridors (ECs) as edges of spatial networks whose nodes are represented by areal units which provide a wide range of ecosystem services (ESs). This methodology detects ECs as important spatial structures aimed at improving the effectiveness of ecological networks by supporting their connection capacity for migration of wild species, their spatial layout and their potential in terms of genetic exchange. EC connection capacity can manifest through minimizing impacts on wild species and genetic flows coming from pressures generated by human activities, such as agriculture and forestry, air and water pollution, gray infrastructure and urban expansion. These threats could cause negative environmental effects as a consequence of the break–up of ecosystem matrices [1].

This study identifies a methodological approach to map ECs and implements such an approach with reference to a network of protected areas located in the spatial context of Sardinia, an Italian insular region. ECs form, together with protected areas, a network representing the spatial framework of regional green infrastructure (GI). Finally, the relation between the ECs and the spatial taxonomy of the landscape components featured by environmental relevance (LCFERs), identified by the Regional Landscape Plan (RLP), is

analyzed, in order to assess if, and to what extent, the present regional spatial zoning code can be used as a basis to implement regulations aimed at protecting ECs.

The conceptual category of connectivity expresses more precisely than that of connection the capacity of connecting ESs, since it includes environmental and landscape aspects, such as the spatial position, the physical continuity, and the presence, type and dimension of natural and anthropic structures, and functional and ecological features, such as the functional perception of species, their ecological and behavioral needs, and their specialization characteristics as well [1–3]. This is in line with Baudry and Merriam [4] who claim that flows of species across ecological networks are often correlated to the connectivity of spatial, mostly linear, elements, which can be defined as ECs.

As per the operational definition of GIs given by the European Commission, spatial connectivity concerning the provision of ESs is strictly related to the conceptual category of ecological network, since a GI can be considered as "[A] strategically planned network of natural and semi-natural areas with other environmental features designed and managed to deliver a wide range of ESs. It incorporates green spaces (or blue if aquatic ecosystems are concerned) and other physical features in terrestrial (including coastal) and marine areas. On land, GI is present in rural and urban settings" [5] (p. 3) and, "The work done over the last 25 years to establish and consolidate the network means that the backbone of the EU's GI is already in place. It is a reservoir of biodiversity that can be drawn upon to repopulate and revitalize degraded environments and catalyze the development of GI. This will also help reduce the fragmentation of the ecosystems, improving the connectivity between sites in the Natura 2000 Network and thus achieving the objectives of Article 10 of the Habitats Directive" [5] (p. 7). This implies that GIs and ESs are strictly related to each other, and that public policies should prioritize ecological networks in terms of environmental protection and enhancement [6]. As a consequence, regional and urban planning processes should adequately manage, improve and monitor the effectiveness of GIs as an ecological network which provides ESs and the spatial connectivity of such systems.

This also entails that GIs are particularly important as in the restoration of biodiversity, the decrease of ecosystem fragmentation and the increase of their capacity of providing ESs [7]. That being so, an operational management goal concerning GIs can be identified as its role in promoting and improving ES provision and habitat restoration [6,8].

The concept of landscape connectivity was introduced by Taylor et al. [9] as a relevant measure of the landscape structure in line with the theory developed by Dunning et al. [10]. According to Taylor et al. [9], landscape connectivity is defined as the "degree to which the landscape facilitates or impedes movement among resource patches" (p. 571). According to With et al. [11], landscape connectivity concerns "the functional relationship among habitat patches, owing to the spatial contagion of habitat and the movement responses of organisms to landscape structure" (p. 151).

In particular, the second definition reflects the dual nature of connectivity, which entails a structural and a functional dimension (structural connectivity, functional connectivity). Structural connectivity is environmentally oriented, while functional connectivity is species-oriented [12]. In this study, the second dimension of connectivity is considered and used. In a nutshell, functional connectivity concerns the movement capacity of species as a function of their intrinsic mobility and of spatial patch suitability to facilitate species movement [9,13].

The concept of landscape connectivity as a means to counter landscape fragmentation has been increasingly embedded into environmental policies, e.g., through technical categories such as greenways, GIs and ECs, in order to address the problem of biodiversity loss [14,15]. The concept of EC is treated in the literature with reference to different scientific and technical profiles (among many, [16–18]). According to Hess and Fisher [18], the use of the term "corridor" is associated with two important theories of conservation biology, i.e., island biogeography [19] and metapopulations [20], which focus on functional connectivity.

Functional connectivity is often analyzed through resistance-based models, where resistance "represents the willingness of an organism to cross a particular environment, the physiological cost of moving through a particular environment, the reduction in survival for the organism moving through a particular environment, or an integration of all these factors" [21] (p. 778). Resistance-based models are widely described and discussed in the literature. The most complex models, such as the circuit theory-based [22] and the individual-based models [23], are difficult to implement due to the overwhelming quantity of input data, and the needed accuracy in data collection and computational power [24]. Building on consolidated approaches available in the current technical and scientific literature [23,25], in this study a least-cost path (LCP) model is defined and implemented in order to identify the spatial structure of ECs.

The article is structured into four sections. In the next section, the study area is described with reference to the protected areas which are assumed as the nodes of the spatial layout of the Sardinian ecological network, and the LCP-based methodology adopted to identify ECs is presented. Moreover, the methodological approach used to analyze the spatial relationship between ECs and LCFERs, identified by the RLP, is described as well.

Section 3 shows the results concerning the identification of Sardinian ECs and the assessment of the relation between ECs and the LCFERs.

Policy implications are discussed in Section 4, whereas future research directions are proposed in the concluding section, with particular reference to the positive aspects and drawbacks of the study.

## 2. Materials and Methods

This section is organized as follows. The first subsection describes the study area and the set of protected areas that are identified as the nodes of the Sardinian regional ecological network. This subsection was written by Lai and reproduced from a previous article by Lai et al. [26]]. The following subsection presents the LCP-based methodological approach implemented to identify the ECs, which is based on studies by Cannas published in a set of articles between 2017 and 2018 [27–30]. Finally, the third subsection discusses the regression model used to assess the relation between ECs and the LCFERs, identified by the RLP. Figure 1 reports a diagrammatic representation of the methodology implemented in this study.

### 2.1. Study Area

Our case study is related to the Sardinian regional context. Sardinia is the second largest Italian island, located in the Western Mediterranean, with an area of around 24,000 km$^2$ [31]. Sardinia is part of the European Mediterranean biogeographical region [32,33].

Two regimes of environmental protection are identified by the Italian legislation, that is, natural protected areas (NPAs) and Natura 2000 sites (N2Ss). In this study, Sardinian NPAs and N2Ss are identified as the Sardinian natural protected sites (NPSs). The set of Sardinian NPSs is shown in Figure 2.

N2Ss are managed by the national government, whereas regional governments rule over the regional NPAs.

Four regional natural parks are established under the provisions of Regional Laws nos. 1999/4, 1999/5, 2014/20 and 2014/21 respectively, that is, Porto Conte, Molentargius-Saline, Gutturu Mannu and Tepilora.

Moreover, our study includes, among the regional NPAs, public woods, permanent oases of faunal protection and Ramsar sites. Public woods, managed by the Regional Agency of Forests, are characterized by significant environmental and landscape values, whose conservation and enhancement are important in order to address and mitigate negative impacts caused by natural disasters, such as fires, floods and landslides. Regional Law no. 1998/23 identifies the permanent oases of faunal protection. Nine Sardinian sites are protected under the provisions of the Ramsar Convention, signed in 1971.

As regards the N2Ss, the Natura 2000 Network includes areas designated under the provisions of Directive no. 92/43/EEC (the Habitats Directive) and Directive no. 2009/147/EC (the Birds Directive), and encompasses more than 27,000 sites, representing the backbone of the European Union's policies on the protection of nature and biodiversity [34]. N2Ss include the following: sites of community interest (SCIs) and special areas of conservation (SACs), established under the Habitats Directive, and special protection areas (SPAs), established under the Birds Directive. SPAs are designated by the European Union member states in relation to a number of scientific criteria, in order to provide bird protection. As regards SCIs and SACs, the designation process develops from Member States' proposals addressed to the European Commission which is responsible for their establishment. SCIs can become SACs within six years of their establishment, provided that conservation measures are identified. Sardinian N2Ss are classified as follows: 31 SPAs, 87 SACs and 10 SCIs [35].

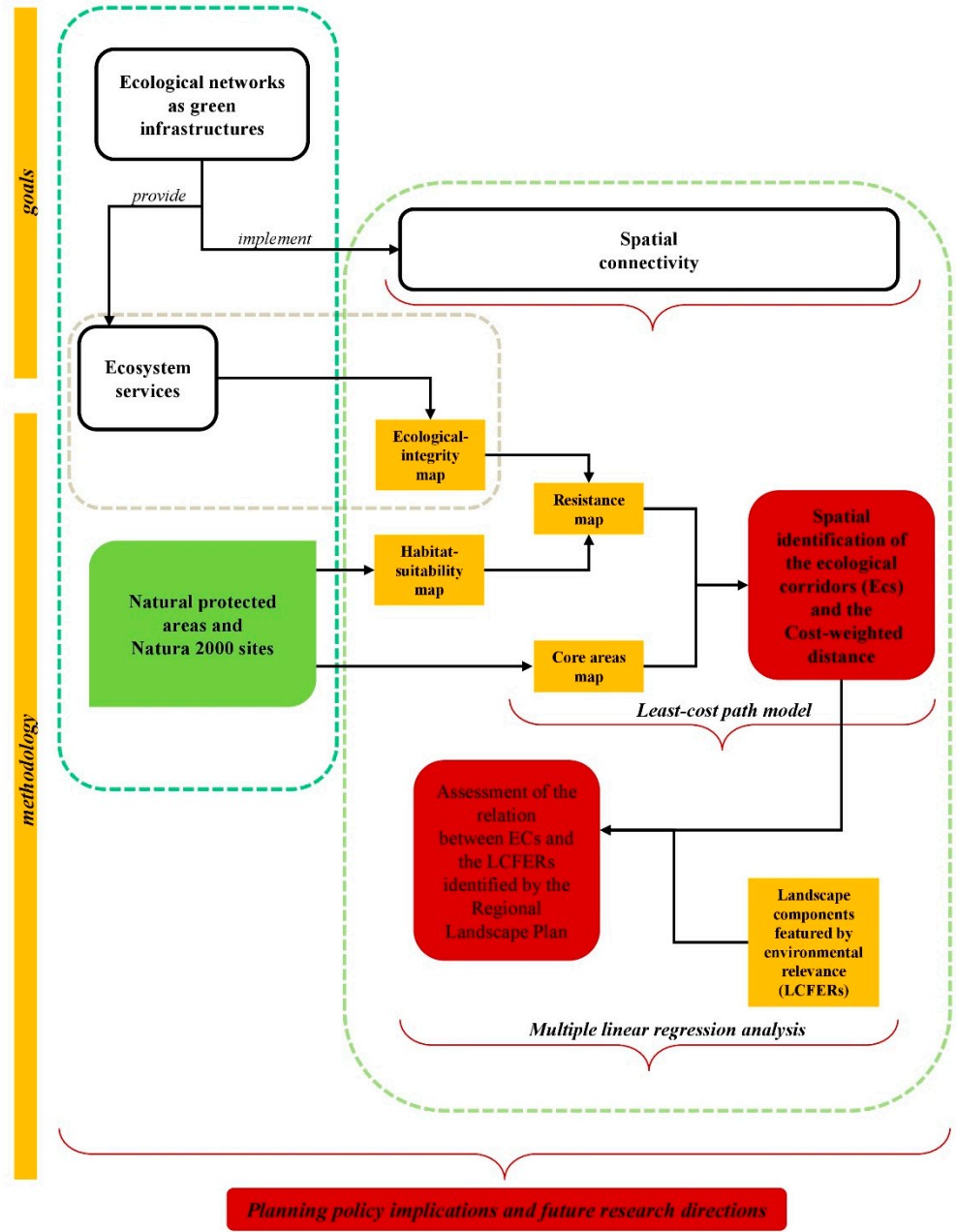

**Figure 1.** The methodological approach.

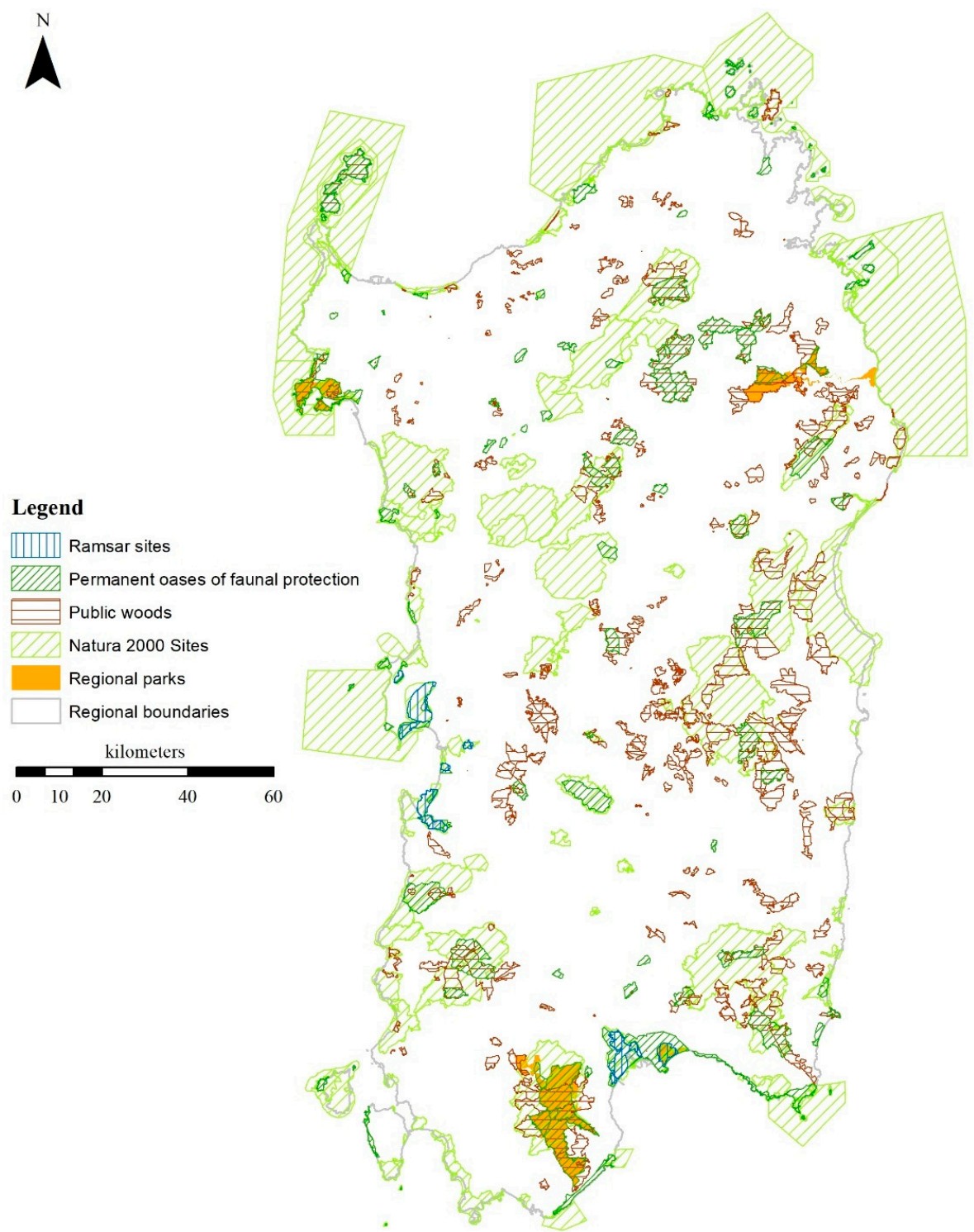

**Figure 2.** The system of the Sardinian protected areas.

## 2.2. Spatial Identification of the Ecological Corridors

LCP models detect spatially identified pathways, which connect habitat patches, characterized by the minimum resistance to species movement, or by the highest probability of movement to take place. LCP models postulate that organisms have an in-depth knowledge of the landscape that leads them to follow the optimal route [14].

According to Sawyer et al. [36], the attractiveness of this typology of models reflects three important points. First, LCP models make it possible to quantitatively compare

potential movement paths within large areas. Secondly, the complex effects of habitats on species movement can be integrated into these models. Finally, LCP models go beyond the limits of analyses based exclusively on structural connectivity by incorporating the species' perception of the surrounding environment. LCP models are particularly effective regarding computational efficiency, model implementation ease, and flexibility related to the inclusion of different environmental profiles and aspects in the model structure [24,37,38]. LCP models often integrate experts' judgments into spatial datasets in order to identify resistance values of areal units [36,39–41].

Building on a methodology developed by Cannas [27–30], the spatial taxonomy of connectivity is identified on the basis of an LCP model, through four phases, as follows:

- definition of a habitat-suitability map;
- definition of an ecological-integrity map;
- definition of a resistance map;
- spatial identification of ECs.

The detail of the input data used in this study is reported in Table 1.

**Table 1.** Description, publication or creation year and source of input data used in this study.

| Data | Description | Year | Source |
|---|---|---|---|
| Sardinian land cover map | Sardinian land cover map is a vector map produced by the Regional Administration of Sardinia, where land covers are classed in relation to four levels. The first three levels report the CLC nomenclature. Linear features include linear entities with a width of less than 25 m, related to roads, railways, and hydrography. As regards polygonal features, the minimum unit mapped is 0.5 hectares within the urban area and 0.75 hectares elsewhere | 2008 | https://www.sardegnageoportale.it/index.php?xsl=2420&s=40&v=9&c=14480&es=6603&na=1&n=100&esp=1&tb=14401 (accessed on 19 April 2022) |
| Species-specific values of habitat suitability | Habitat suitability species-specific values are defined within a study commissioned by the Regional Administration of Sardinia to AGRISTUDIO et al. [42]. The values concern species and habitats of community interest within the Sardinian N2Ss. The study provides habitat-suitability species-specific values, on an ordinal scale between 0 and 3 (0: non–suitable; 3: extremely suitable), for each CLC class of the Sardinian land cover map in relation to each Sardinian N2S. The evaluation is based on experts' judgments | 2011 | Unpublished work |
| Values of ecological integrity | Ecological integrity values are developed by Burkhard et al. [43,44] in relation to each of the 44 third-level land cover classes of the CLC taxonomy through experts' judgments. The ecological-integrity index is equal to the sum of the scores associated to seven ES-supply indicators (abiotic heterogeneity, biodiversity, biotic waterflows, metabolic efficiency, energy capture, reduction of nutrient loss and storage capacity) | 2009 | https://landscape-online.org/index.php/lo/article/view/LO.200915/67 (accessed on 19 April 2022) |

**Table 1.** *Cont.*

| Data | Description | Year | Source |
|---|---|---|---|
| Map of core areas | The map of core areas is a vector map, developed by the authors which combines different typologies of protected areas: national parks (NPs), regional parks (RPs), public woods (PWs), permanent oases of faunal protection (POFPs), Ramsar sites (RSs) and N2Ss | 2009 as for NPs | https://webgis2.regione.sardegna.it/geonetwork/srv/ita/catalog.search#/metadata/R_SARDEG:YDBMD (accessed on 19 April 2022) |
| | | 2013 as for RPs and RSs | https://webgis2.regione.sardegna.it/geonetwork/srv/ita/catalog.search#/metadata/R_SARDEG:585dc615-71d2-4318-ade6-6b3341781987 (accessed on 19 April 2022) |
| | | 2009 as for PWs | https://webgis2.regione.sardegna.it/geonetwork/srv/ita/catalog.search#/metadata/R_SARDEG:BLFQZ (accessed on 19 April 2022) |
| | | 2005 as for POFPs | https://webgis2.regione.sardegna.it/geonetwork/srv/ita/catalog.search#/metadata/R_SARDEG:DSDPP (accessed on 19 April 2022) |
| | | 2013 as for RSs | https://webgis2.regione.sardegna.it/geonetwork/srv/ita/catalog.search#/metadata/R_SARDEG:f52f111d-2a2e-4870-a623-6d6f11dc4f1d (accessed on 19 April 2022) |
| | | 2021 as for N2Ss | https://www.eea.europa.eu/data-and-maps/data/natura-13/natura-2000-spatial-data/natura-2000-shapefile-1 (accessed on 19 April 2022) |
| Landscape components featured by environmental relevance | The LCFER map is a vector map developed by the Regional Administration of Sardinia in relation to the RLP implementation code. As explained in Section 2.3, the LCFER map classifies the regional land into three typologies of areas: natural and subnatural, seminatural, and agricultural and forestry | 2005 | https://webgis2.regione.sardegna.it/geonetwork/srv/ita/catalog.search#/metadata/R_SARDEG:BYBET (accessed on 19 April 2022) |

The first phase aims at defining a habitat-suitability map, where habitat suitability is defined as the probability of habitat use by species. The elaboration of this map is based on the Sardinian land cover vector map and on a study concerning species-specific values of habitat suitability. Land covers are classed according to the Sardinian land cover vector map produced by the Regional Administration of Sardinia in 2008, at the third level of the CORINE Land Cover (CLC) nomenclature. Moreover, species-specific values of habitat suitability are identified on the basis of a study by AGRISTUDIO et al. [42], commissioned by the Regional Administration of Sardinia, concerning the conservation status of species and habitats of community interest within the Sardinian N2Ss. The study provides habitat-suitability species-specific values, on an ordinal scale between 0 and 3 (0: non–suitable; 3: extremely suitable), for each CLC class of the Sardinian land cover map in relation to each Sardinian N2S. The evaluation is based on experts' judgments. A habitat-suitability map is elaborated on the basis of two assumptions. First, the habitat suitability species-specific values, associated with land cover classes located in the N2Ss by the AGRISTUDIO et al.'s [42] study, are associated with the same land cover classes of areas outside the N2Ss as well. Secondly, the total value of the species-specific habitat suitability associated with each land cover class is equal to the average value of the single species-specific values associated with the land cover class. Finally, a habitat-suitability vector map is defined, which identifies a taxonomy concerning the entire regional area.

The second phase aims at defining an ecological-integrity map, which builds on studies developed by Burkhard et al. [43,44], where an assessment of land cover classes' capacities to provide ESs is implemented through experts' judgments, on the basis of the founding concept that the higher the ecological integrity, the higher the suitability to species' transition and movement. Ecological integrity concerns supporting ESs defined as ESs which help to maintain and enhance the supply of the other types of ES, namely provisioning, regulating and cultural ES. The ecological-integrity index is equal to the sum of the scores associated with seven ES supply indicators (abiotic heterogeneity, biodiversity, biotic waterflows, metabolic efficiency, exergy capture, reduction of nutrient loss and storage capacity) that represent supporting ESs in relation to each of the 44 third-level land cover classes of the CLC taxonomy. As a result, by mapping the values of the ecological-integrity index, an ecological-integrity vector map is obtained for the entire regional area.

The third phase aims at defining the resistance map by means of the habitat-suitability and ecological-integrity maps, building on a study by LaRue and Nielsen [45]. First, the two vector maps are converted into raster maps; secondly, two maps are defined by mapping the inverse of the sum of the habitat suitability and of the ecological-integrity index; thirdly the new raster maps are scaled, on an ordinal scale between 1 and 100 (1: the lowest resistance; 100: the highest resistance), according to a study by the European Environment Agency [8]. Finally, the values of the two rescaled raster maps are summed-up and mapped on a patch-by-patch basis. The resulting spatial taxonomy is the resistance map.

The fourth phase aims at spatially identifying ECs that connect the Sardinian NPSs through the use of the Linkage Pathways Tool (LPT) of the GIS Linkage Mapper (LM) Toolbox. LPT implements the LCP approach by identifying the Cost-Weighted Distance (CWD) [46]. The LCP laying between two core areas is identified by the path which shows the minimum CWD. Input data required by the LPT are a vector map of core areas and a raster resistance map. In this study, each core area is identified either by a single NPS, in case the overlapping of multiple NPSs does not occur, or by the spatial envelope of overlapping NPSs, whereas phases 1 thru 3 identify the resistance map.

The CWD of a path between two core areas is obtained by: i. averaging the resistance values of pairs of adjacent patches; ii. multiplying such average values times the geometric distance of the patches' centers [47]; and, iii. Summing up the results of item ii. along the path.

The relevant outputs offered by LPT are the linear developments of the ECs and the raster map of the CWD values. Figure 3 shows the implementation of the LPT processing process.

LPT proceeds as follows, in order to identify the LCP between two core areas A and B.

First, the normalized distance related to each patch i connecting A and B, $ND_{iAB}$, is calculated, as follows:

$$ND_{iAB} = CWD_{iA} + CWD_{iB} - LCWD_{AB}, \qquad (1)$$

where: $ND_{iAB}$ is the normalized distance between A and B measured along a path which includes patch i; $CWD_{iA}$ and $CWD_{iB}$ are the cost-weighted distances from patch i to core areas A and B; and, $LCWD_{AB}$ is the least CWD, i.e., the CWD measured along the LCP connecting A and B [46].

Secondly, the LCP, i.e., the EC, connecting A and B, is identified by the spatial sequence of patches j's which show $ND_{iAB} = 0$.

### 2.3. Relation between ECs and Landscape Components

The LCFERs represent a spatial taxonomy of the regional land aimed at defining differentiated levels of protection depending on the value of nature and natural resources. This taxonomy was defined in the RLP approved by the Deliberation of the Sardinian Regional Government no. 36/7 of 5 September 2006, and implemented a protection regime which did not take account of ecological corridors, whereas their importance was recognized by art. 10 of the Habitats and Birds Directives, according to which ECs make the Natura 2000 Network internally connected from the functional and ecological points

of view. As a consequence, ECs can be considered areal structures connecting habitats to enhance and support biodiversity, and, in so doing, increase the ES provision [29,30].

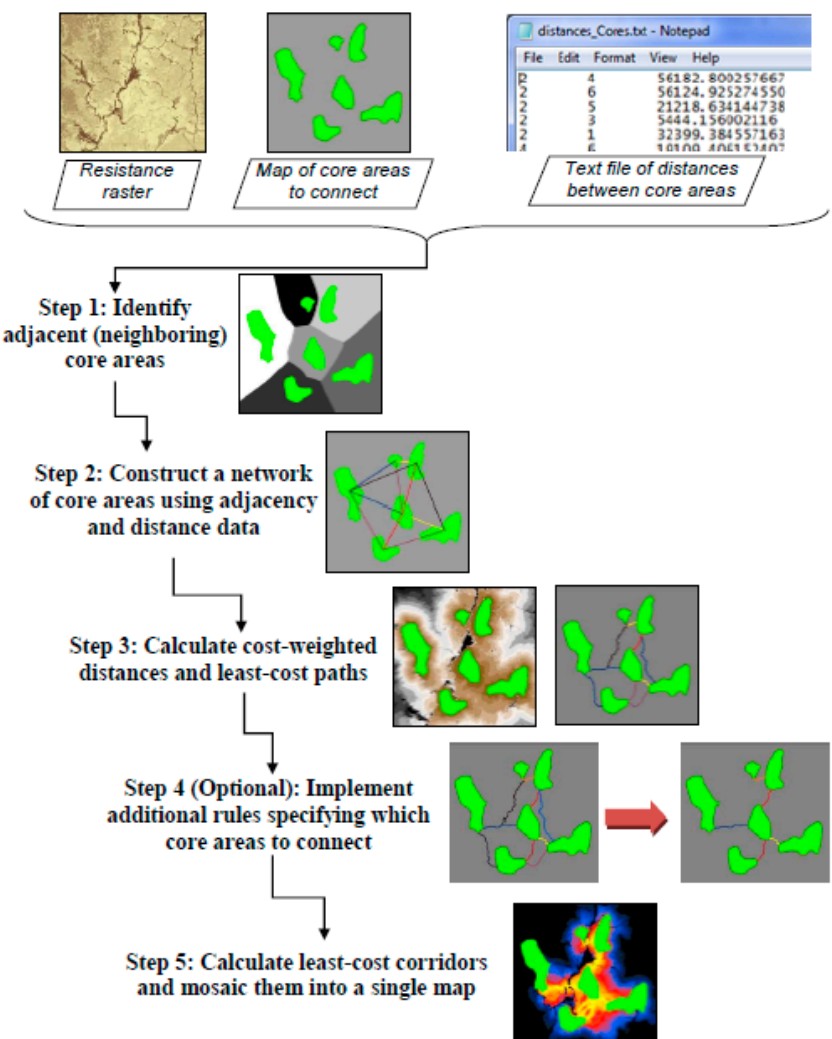

**Figure 3.** Processing process of LPT. Source: McRae and Kavanagh [46] (p. 11).

Thus, the implementation of EC protection into the Sardinian spatial planning framework, established under the provisions of the RLP code, has to be developed by identifying ECs as areas with the highest protection level among the LCFERs.

The spatial layout of ECs connecting core areas is defined by the raster map of CWD values clustered into ten deciles, whose second upper limit is assumed as the threshold for the inclusion of a patch in an EC [27]. The CWD of a patch j, included in an EC connecting the core areas A and B, is calculated as follows:

$$CWD_j = CWD_{jA} + CWD_{jB}, \qquad (2)$$

where $CWD_{jA}$ and $CWD_{jB}$ are the cost-weighted distances from patch j to core areas A and B.

The assessment of the relations between ECs and LCFERs is implemented through a linear regression model which relates the eligibility of a patch to be included in an EC and the areas of the LCFERs overlaid by the corridors.

The LCFERs classed by the RLP implementation code (IC) are the following:

- natural and subnatural areas, which include: scrub vegetation in dry areas and wetlands (areas covered with sparse vegetation, between 5% and 40%; riparian areas covered with non–arboreal vegetation; Mediterranean scrub; river beds larger than

25 m; inland marshes; salt marshes; rock faces); and, woodlands (mixed coniferous and broadleaf woods; broadleaf woods);

- seminatural areas, which include: grasslands (steady meadows; natural pastures; thickets and shrublands; garrigues; natural recolonization areas); and, cork and chestnut woods;
- areas dedicated to agriculture and forestry, which include: specialized and tree crops (vineyards; orchards; temporary olive– and vineyard–related crops; temporary crops related to other permanent crops); artificial woods (coniferous woods; poplar, willow and eucalypt woods; other trees for timber; arboriculture with coniferous forest trees; artificial recolonization areas); and, specialized herbaceous crops, agricultural and forest areas, and uncultivated areas (non–irrigated arable land; artificial meadows; simple arable land and full–field horticultural crops; paddies; breeding grounds; greenhouse crops; complex parcel cropping systems; areas characterized by prevailing agricultural crops and residual important natural land; uncultivated areas).

According to the RLP IC, the protection regime concerning natural and subnatural areas forbids whichever spatial transformation, including new buildings or land use modifications, which is likely to undermine the ecosystem structure, steadiness and functionality, or the landscape enjoyment potential. As for dunal and retrodunal habitats featured by non–arboreal vegetation or Mediterranean scrub, vehicle and pedestrian access and temporary installations are not allowed if they may put at risk natural resources conservation. Moreover, the RLP IC forbids the implementation of spatial transformations which may cause water pollution or landfill as regards wetlands. Finally, afforestation is not allowed if potentially harmful to priority habitats designed by the Habitats and Birds Directives, with the exception of conservation operations.

As for seminatural areas, the RLP IC states that whichever spatial transformation, including new buildings or land use modifications, which is likely to undermine the ecosystem structure, steadiness and functionality, or the landscape enjoyment potential, is not allowed, with the exception of operations aimed at improving the ecosystems structure and functioning, the conservation status of biotic and abiotic natural resources, and at mitigating environmental hazard and degradation of natural resources. In woodlands, land-use modifications are forbidden except for land-use changes related to the development of new faunistic or floristic populations and to the enhancement of the habitats of protected wildlife. Moreover, new facilities are not permitted, whereas restoration of existing buildings is allowed provided that they will be used to improve the conditions of nature and natural resources, and that the operations do not entail an increase in building volume, floor area and covered surface.

New infrastructure, such as roads, power lines, hydraulic pipelines, etc., which may alter the forest land cover or increase fire or pollution hazards are not allowed in seminatural areas, with the exception of operations aimed at forest management and soil protection. Furthermore, the RLP IC forbids new roads, power lines and wind turbines close to wetlands and to areas characterized by the presence of species of community interest, especially with reference to birdlife, which may generate negative impacts on the landscape perception. River systems and riparian areas have to be protected from soil–sealing operations, afforestation implemented by using alien species and removal of sand and sediments from the river beds.

As for dunal systems and sandy seashores, vehicle traffic is strictly forbidden, and sand and sediment removal are not allowed as well. Finally, a general rule concerning seminatural areas concerns a ban on the use of alien species for afforestation, reforestation, and renaturation.

With reference to areas dedicated to agriculture and forestry, and uncultivated areas, the RLP IC forbids transitions from agriculture and forestry to other land uses, with the exception of changes motivated by reasons related to the implementation of relevant public utilities for which it is demonstrated that no other location is presently available. Limited land-use transitions are allowed to make more effective infrastructure, facilities and machinery exclusively devoted to agriculture or forestry. Moreover, the biodiversity

improvement as regards native species of agrarian interest, the conservation of local traditional agricultural systems and the protection of typical rural scenery are indicated as important addresses, stated as planning rules as per art. 29 of the RLP IC, in particular with reference to periurban zones and historic terrace farming areas.

All in all, the RLP IC identifies rules concerning natural, subnatural and seminatural areas which are almost entirely consistent with a nature protection regime aimed at strengthening the effectiveness of ECs. The main regulatory feature of the RLP IC with respect to these areas is the general objective of protecting the structure and functionality of ecosystems, biodiversity, nature and natural resources, with particular attention to habitats and species identified by the Habitats and the Birds Directives, dunal and coastal environments, and wetlands as main sources of biodiversity, especially as regards birdlife. In woodlands, modifications of land use are not allowed, except for the improvement of wildlife habitats and an increase in faunistic and floristic populations. Rules concerning agriculture and forestry are less restrictive since land-use transitions are allowed if they aim at improving farm and forest productivity, even though protection of traditional practices, scenery and biodiversity protection with reference to rural landscapes and environments are targeted as important planning policy goals.

The relation between ECs and the LCFERs described so far is analyzed through a multiple linear regression model which assesses the correlations between CWD and the areas of the LCFERs which overlay ECs. The model takes the following form:

$$ECWD = \beta_0 + \beta_1 SCRB + \beta_2 WOOD + \beta_3 GRAS + \beta_4 CCHW + \beta_5 SPTC + \beta_6 ARWO + \beta_7 HAFU + \beta_8 ALTD, \quad (3)$$

where dependent and explanatory variables identify the areal dimensions of ECs and of the overlays of ECs and the LCFERs:

- ECWD is the CWD of a patch included in an EC;
- SCRB is for scrub vegetation in dry areas and wetlands;
- WOOD is for woodlands;
- GRAS is for grasslands;
- CCHW is for cork and chestnut woods;
- SPTC is for specialized and tree crops;
- ARWO is for artificial woods;
- HAFU is for specialized herbaceous crops, agricultural and forest areas, and uncultivated areas;
- ALTD is a control variable that represents the average altitude in an EC.

The outcomes of the regression model identify the quantitative correlations between the linear dimension of ECs, ECWD, and the presence of LCFERs.

As per many studies related to correlations between spatial variables, a regression model is used since no prior hypothesis seems to be plausible as regards the effect of covariates on the dependent variable (among many: [48–51]).

Thus, a surface, characterized by an unknown equation, representing a spatial phenomenon featured by n factors, is approximated, in an infinitesimal neighborhood of one of its points, by its tangential hyperplane. The infinitesimal area shared by the hyperplane and the surface is identified by the known equation of the tangential hyperplane, that is, by the linear relation between the covariates. Such linear relation locally approximates the unknown surface. That being so, the multiple regression model (3) estimates the trace of an eight-dimensional hyperplane on an eight-dimensional surface whose equation is unknown [52,53], which shows the linear correlations between ECWD and the eight dependent variables defined above.

The variable ALTD is utilized as a control variable to check the effect of the altitude of an EC on its areal dimension; so, if the estimate of the coefficient $\beta_8$ were significant, this would imply that the altitude is likely to cause a relevant impact on ECWD. The sign of the estimated coefficient indicates if the impact is positive or negative, i.e., if the greater the altitude, the lower ECWD, or the other way around.

Finally, a 5% *p*-value significance test is used with reference to the estimated coefficients of model (3) to see if their estimates are significantly different than zero.

## 3. Results

This section is organized as follows. The first subsection presents the spatial layout of the ECs identified through the implementation of the methodology described in Section 2.2. The following subsection operationalizes the regression model defined in Section 2.3.

### 3.1. The Spatial Layout of Ecological Corridors

The implementation of the methodological approach developed by Cannas [27–30], and described in Section 2.2, is developed through four phases, which each generates one or more outputs necessary to carry out the following phase.

The first phase provides a habitat-suitability map (see Figure 4), where habitat suitability species-specific values range from 0.1 to 1.65.

The second phase produces an ecological-integrity map (see Figure 5), where ecological integrity values range from 0.1 to 32.

The third phase delivers a resistance map (see Figure 6), where resistance values range from 2 to 200.

The last phase generates two outputs: i. the raster map of the CWD values; ii. the spatial identification of the ECs that connect the NPSs of the Sardinian protected area network. Figure 7 shows the ECs identified in the study area and Figure 8 reports the CWD values, included in a range between 0 to 225,201 km. As described in Section 2.3, the CWD values are clustered into ten deciles, whose second upper limit is assumed as the threshold for the inclusion of a patch in an EC. The CWD values included in the first two deciles range from 0 to 9741 km. In Figures 7 and 8, the ECs are shown as linear elements.

Through LPT, 240 ECs are identified, with CWD ranging between 0.07 km and 27.34 km. Moreover, two important qualitative attributes of the ECs connecting two core areas have to be emphasized: the ratio of the CWD to the Euclidean distance (CWD/ED) and the ratio of CWD to the length of the EC (CWD/LCP) [54,55]. The former measures the resistance to species movement between two core areas in relation to their proximity, i.e., the connectivity quality of the connecting EC, as long as the latter identifies the average resistance to species movement along with the EC which connects two core areas.

With reference to the CWD/ED index, ECs nos. 22, 112, and 122 show the lowest values and, as a consequence, the highest connectivity quality (see Table 2 and Figure 9), whereas ECs nos. 12, 228, and 9 show the highest values and, that being so, the lowest connectivity quality (see Table 2 and Figure 10).

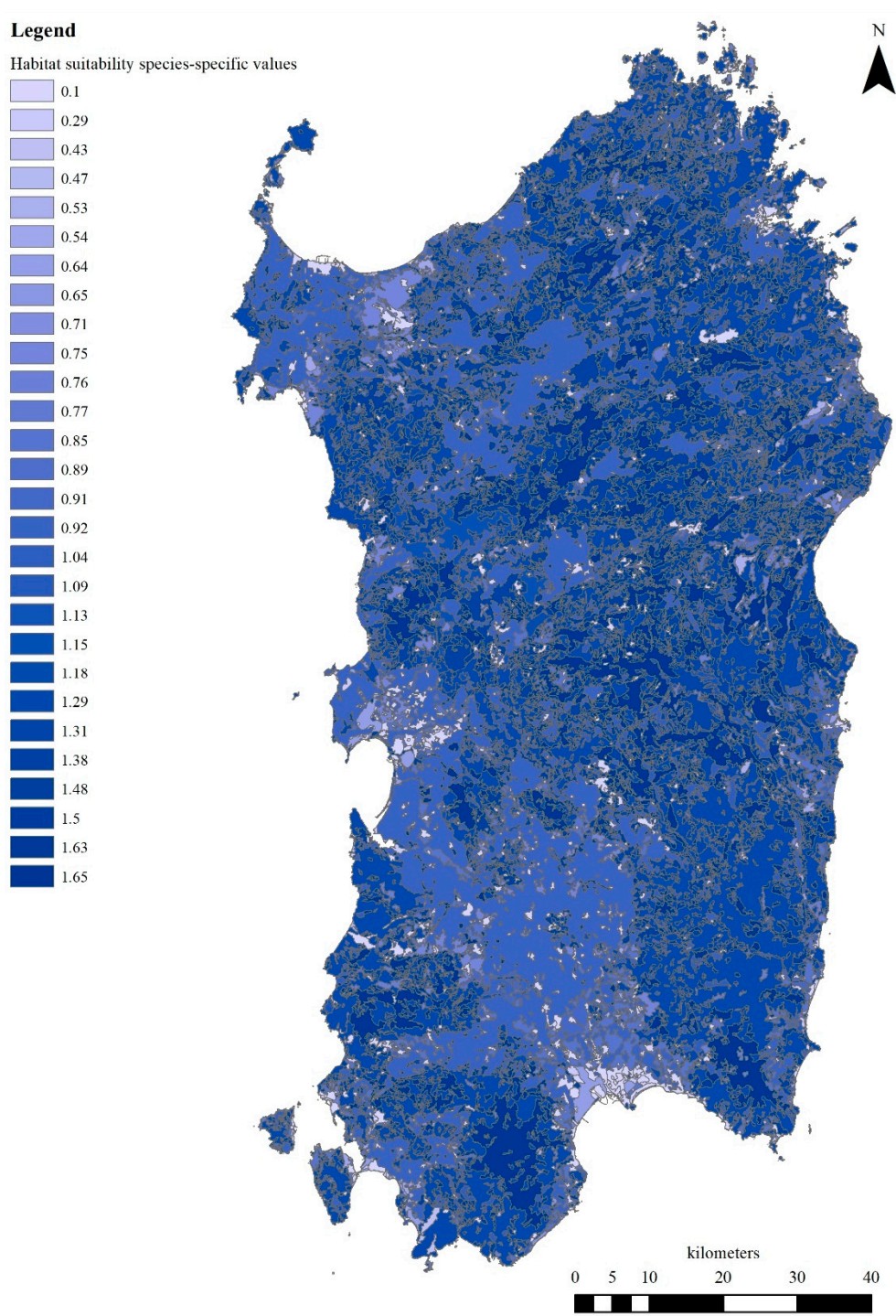

**Figure 4.** Habitat-suitability map.

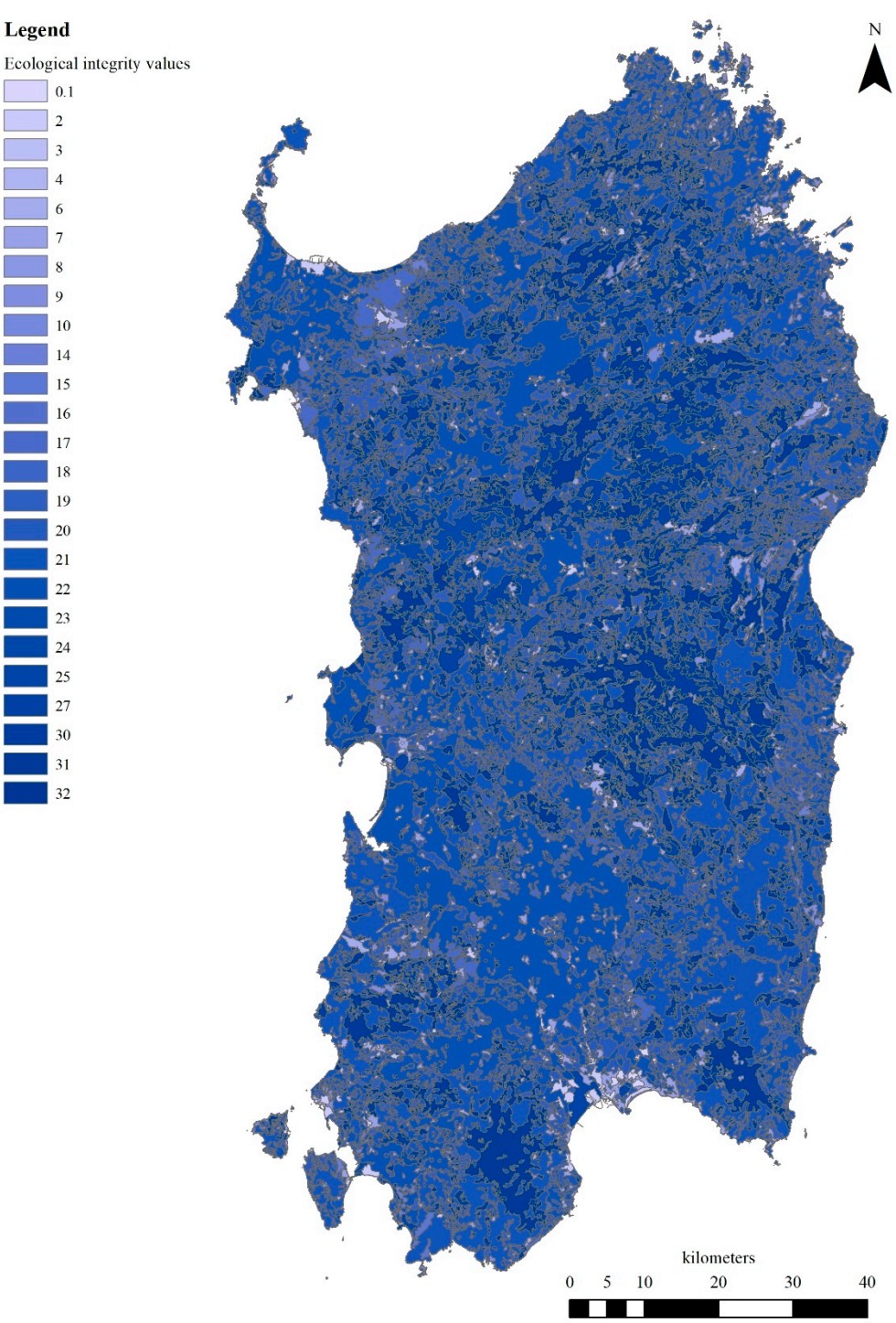

**Figure 5.** Ecological-integrity map.

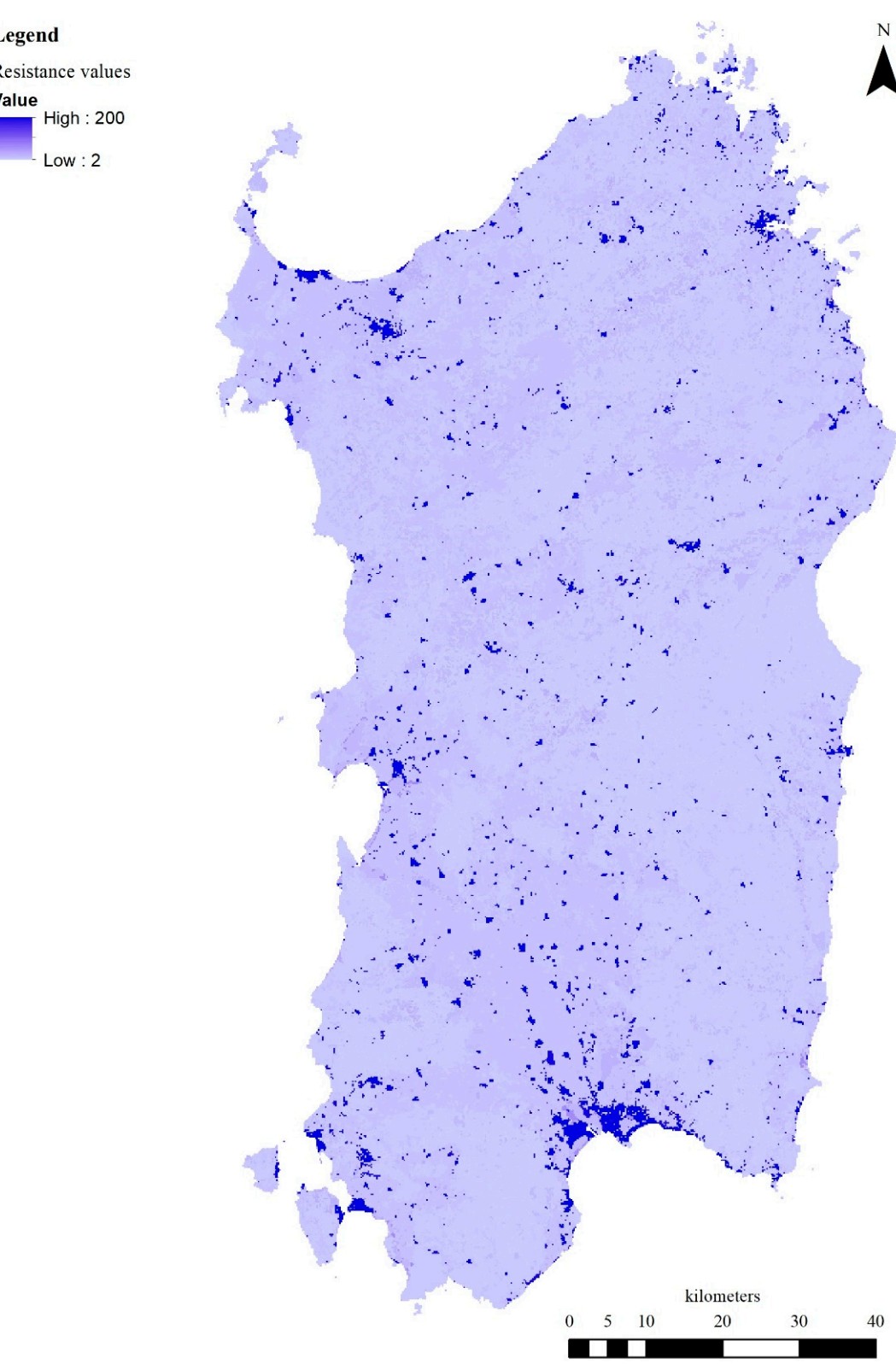

**Figure 6.** Resistance map.

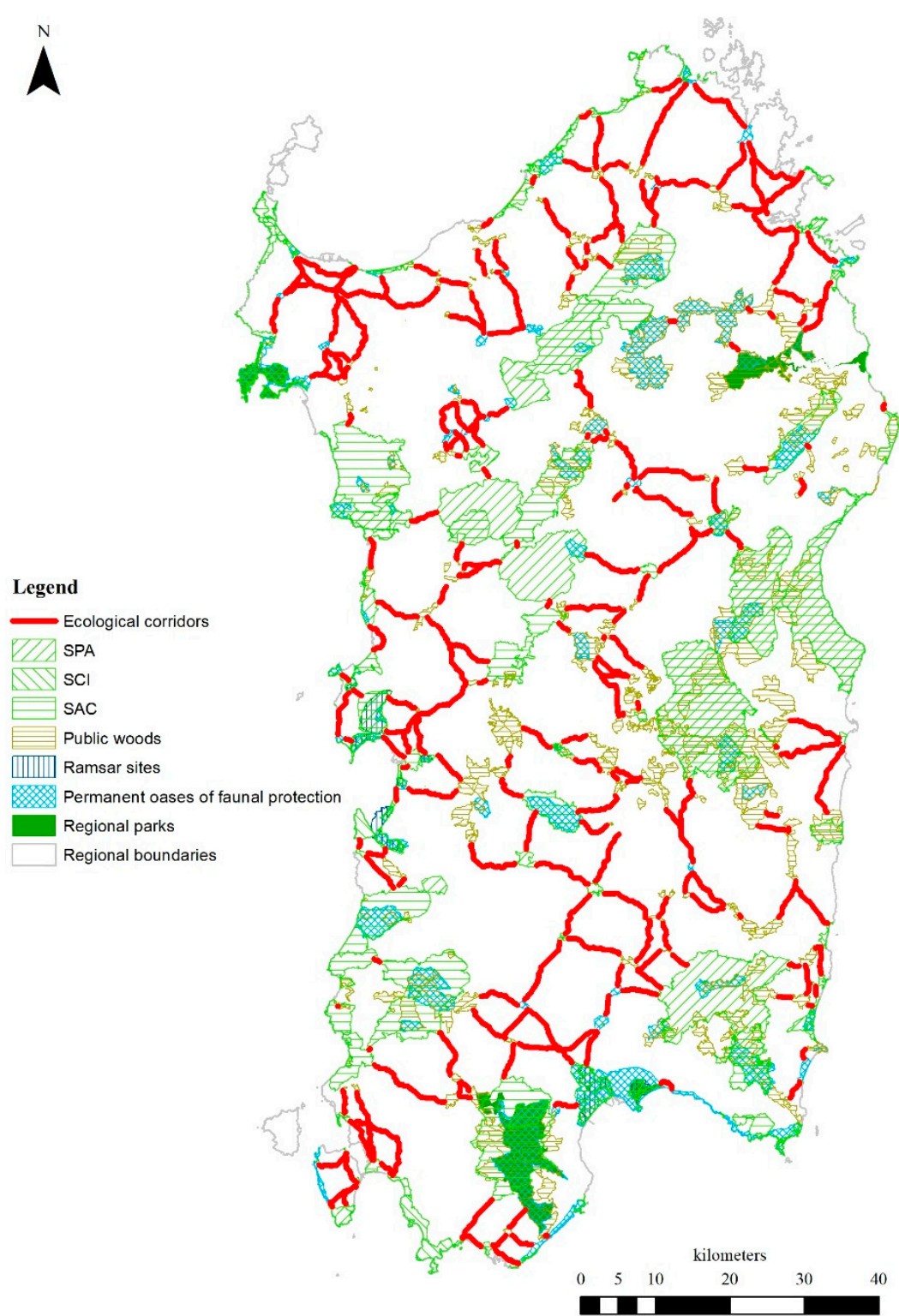

**Figure 7.** Ecological corridors connecting the NPSs.

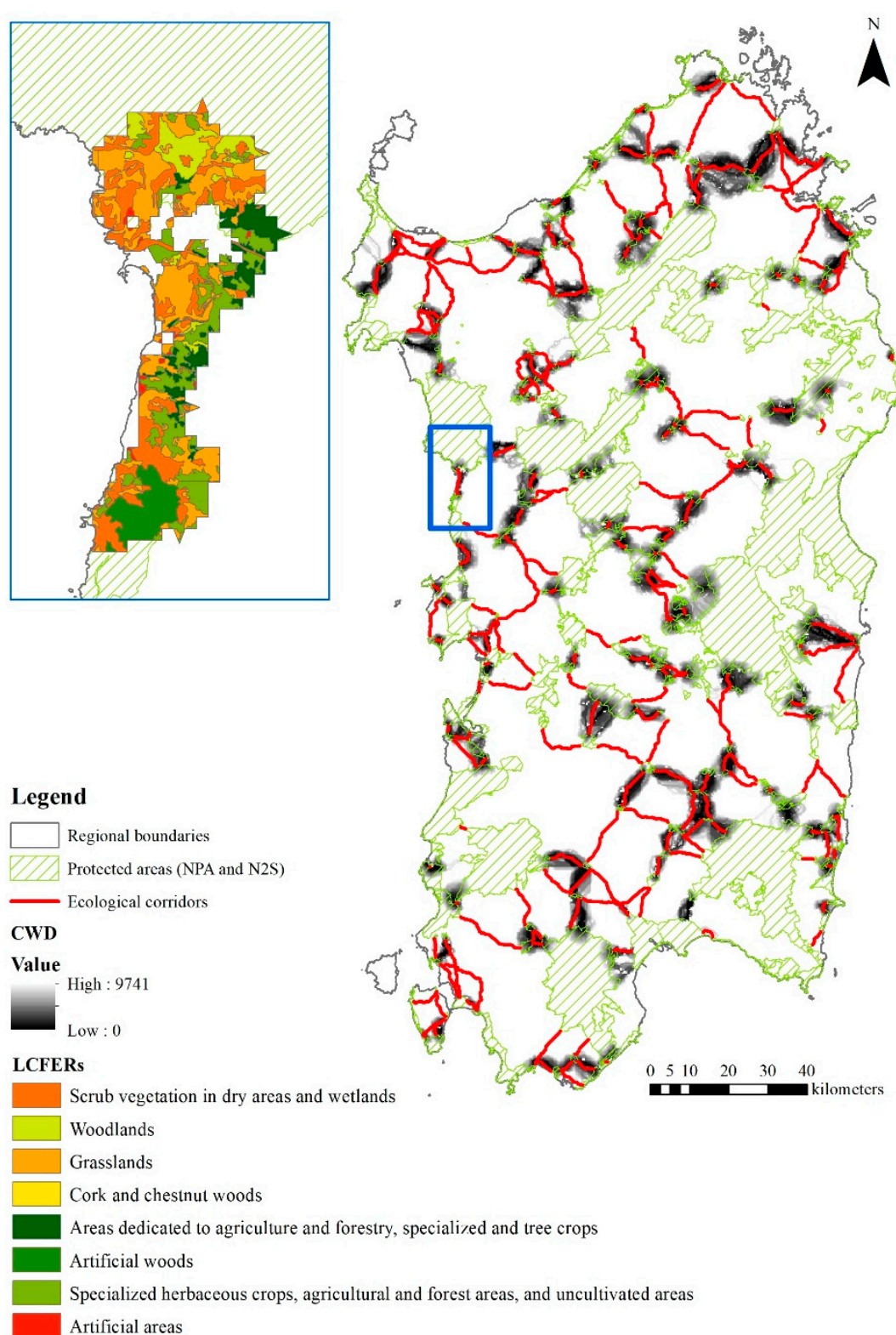

**Legend**

☐ Regional boundaries

▨ Protected areas (NPA and N2S)

▬ Ecological corridors

**CWD**

**Value**

High : 9741

Low : 0

**LCFERs**

■ Scrub vegetation in dry areas and wetlands

■ Woodlands

■ Grasslands

■ Cork and chestnut woods

■ Areas dedicated to agriculture and forestry, specialized and tree crops

■ Artificial woods

■ Specialized herbaceous crops, agricultural and forest areas, and uncultivated areas

■ Artificial areas

**Figure 8.** Identification of ecological corridors and of CWD values included in the second's upper limit decile (map on the **right**), and the overlapping map of CWD values and the LCFERs (**upper-left** map).

**Table 2.** Name and typology of NPSs included within core areas connected by EC which shows the highest and lowest values of CWD/ED index and CWD/LCP index.

| EC Code | Core Area Code | Name of Connected NPSs | Typology of NPSs |
|---|---|---|---|
| 22 | 7 | Monte dei Sette Fratelli | SPA |
| | | Monte dei Sette Fratelli e Sarrabus | SAC |
| | | Monte Genis | Permanent oases of faunal protection |
| | | Castiadas-Sette Fratelli | Permanent oases of faunal protection |
| | | Campidano | Permanent oases of faunal protection |
| | | Campidano | Public woods |
| | | Campidano Santo Barzolu | Public woods |
| | | Castiadas | Public woods |
| | | San Vito | Public woods |
| | | Sa Scova | Public woods |
| | | Sette Fratelli | Public woods |
| | | Villasalto | Public woods |
| | 24 | Baccu Arrodas—Rio Molas | Public woods |
| 122 | 47 | Olzai | Public woods |
| | 148 | Monte Gonare | SAC |
| 112 | 49 | Ussai | Permanent oases of faunal protection |
| | | Barigadu | Public woods |
| | 40 | Foresta di Uatzo | Public woods |
| 12 | 4 | Parco Naturale Regionale di Molentargius saline | Natural regional park |
| | | Monte Sant'Elia, Cala Mosca e Cala Fighera | SAC |
| | | Stagno di Cagliari, Saline di Macchiareddu, Laguna di Santa Gilla | SAC |
| | | Stagno di Molentargius e territori limitrofi | SAC |
| | | Torre del Poetto | SAC |
| | | Stagno di Cagliari | SPA |
| | | Saline di Molenatrgius | SPA |
| | | Santa Gilla | Permanent oases of faunal protection |
| | | Stagni di Quartu Molentargius | Permanent oases of faunal protection |
| | | Stagno di Molentargius | Ramsar Site |
| | | Stagno di Cagliari | Ramsar Site |
| | 10 | Bruncu de Su Monte Moru—Geremean (Mari Pintau) | SAC |
| | | Costa di Cagliari | SAC |
| | | Capo Carbonara e stagno di Notteri—Punta Molentis | SPA |
| | | Fascia litoranea sud orientale | Permanent oases of faunal protection |

**Table 2.** *Cont.*

| EC Code | Core Area Code | Name of Connected NPSs | Typology of NPSs |
|---|---|---|---|
| 228 | 140 | Stagno di Santa Caterina | SAC |
| | 152 | Stagno di Porto Botte | SAC |
| | | Isola Rossa e Capo Teulada | SCI |
| | | Promontorio, dune e zona umida di Porto Pino | SCI |
| 9 | 3 | Sassu-Cirras | SAC |
| | | Stagno di S'Ena Arrubia e territori limitrofi | SAC |
| | | Stagno di S'Ena Arrubia | SPA |
| | | S'Ena Arrubia | Permanent oases of faunal protection |
| | | S'Ena Arrubia | Ramsar Site |
| | 5 | Stagno di Pauli Maiori di Oristano | SAC |
| | | Stagno di Santa Giusta | SAC |
| | | Stagno di Pauli Maiori | SPA |
| | | Pauli Maiori | Permanent oases of faunal protection |
| | | Stagno di Pauli Maiori | Ramsar Site |
| 192 | 80 | Altopiano di Campeda | SAC |
| | | Catena del Marghine e del Goceano | SAC |
| | | Piana di Semestene, Bonorva, Macomer e Bortigali | SPA |
| | | Monte Pisanu | Permanent oases of faunal protection |
| | | Foresta Anela | Permanent oases of faunal protection |
| | | Anela | Public woods |
| | | Bono | Public woods |
| | | Monte Artu | Public woods |
| | | Monte Bassu | Public woods |
| | | Monte Burghesu | Public woods |
| | | Monte Pisanu | Public woods |
| | 81 | Foresta Fiorentini | Permanent oases of faunal protection |
| | | Fiorentini | Public woods |
| | | Monte Pirastru | Public woods |
| 119 | 43 | Pabarile | Public woods |
| | 142 | Riu Sos Mulinos—Sos Lavros—M. Urtigu | SAC |

**Table 2.** *Cont.*

| EC Code | Core Area Code | Name of Connected NPSs | Typology of NPSs |
|---|---|---|---|
| 14 | 4 | Parco Naturale Regionale di Molentargius saline | Natural regional park |
| | | Monte Sant'Elia, Cala Mosca e Cala Fighera | SAC |
| | | Stagno di Cagliari, Saline di Macchiareddu, Laguna di Santa Gilla | SAC |
| | | Stagno di Molentargius e territori limitrofi | SAC |
| | | Torre del Poetto | SAC |
| | | Stagno di Cagliari | SPA |
| | | Saline di Molenatrgius | SPA |
| | | Santa Gilla | Permanent oases of faunal protection |
| | | Stagni di Quartu Molentargius | Permanent oases of faunal protection |
| | | Stagno di Molentargius | Ramsar Site |
| | | Stagno di Cagliari | Ramsar Site |
| | 15 | Ovile Sardo | Permanent oases of faunal protection |

As regards the CWD/LCP index, ECs nos. 192, 119, and 112 show the lowest values and, as a result (see Table 2 and Figure 11), the lowest average resistance to species movement along the path, while ECs nos. 12, 228, and 14 show the highest values and, for that reason, the highest average resistance to species movement (see Table 2 and Figure 12).

*3.2. Discussion on the Overlay of Ecological Corridors and Landscape Components*

The estimated coefficients of the explanatory variables of model (3) show the correlations between the ECWD of a parcel included in an EC and the covariates of the multiple linear regression, identified by the LCFERs and by the control variable ALTD.

The descriptive statistics related to dependent and explanatory variables of model (3) are shown in Table 3, whereas Table 4 reports the estimates of the multiple linear regression.

The estimated coefficient of the altitude-related variable shows significant *p*-values and a positive sign. This implies that a decrease in ECWD is associated with lower altitudes, everything else being equal, which is entirely consistent with expectations, since higher connectivity, or lower ECWD, is expected to take place in flat areas, generally characterized by comparative lower altitudes. Our findings entail that a decrease of 100 m in altitude will be correlated to a decrease of about 145 m in ECWD.

Since the estimate of the coefficient of the control variable is statistically significant and consistent with expectations, the estimated effects on ECWD generated by the covariates related to the LCFERs can be considered reliable as regards the implementation of model (3).

The results of the coefficient estimate of model (3), reported in Table 4, are related to the explanatory variables expressed by the percentage share of the area of a landscape component in the total area of a patch. Such estimates show the marginal effects of the explanatory variables on ECWD. The estimates exhibit *p*-values lower than 6.6%, with the exception of scrub vegetation in dry areas and wetlands (SCRB), which, at any rate, shows a weakly significant *p*-value (10.8%). The comprehensive goodness of fit is also endorsed by the value of the adjusted correlation coefficient, which exceeds 70%.

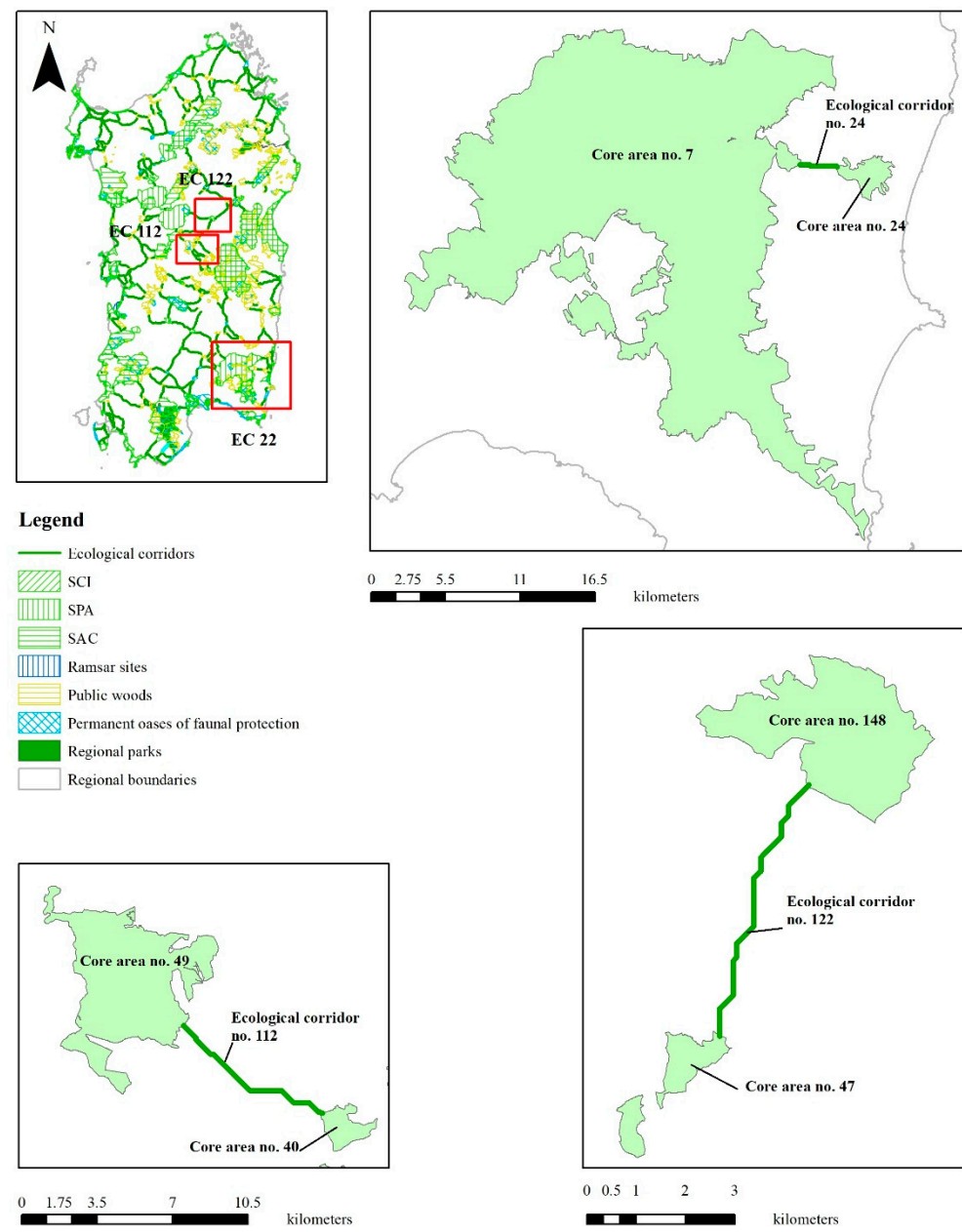

The codes of the core areas are identified in tab. 1

**Figure 9.** Spatial identification of the ECs which show the lowest values of the CWD/ED index.

Moreover, the regression results put in evidence that all the LCFERs are correlated to increases in the eligibility of a patch to be included in an EC, i.e., an increase in the percentage area of an LCFER is correlated to a decrease in ECWD everything else being equal, except for specialized and tree crops (SPTC), whose coefficient is positive and indicates that an average increase of 1% in SPTC is associated to an average increase of 7.7 m in the CWD of ECs.

As for the other LCFERs, the outcomes show that woodlands (WOOD) and cork and chestnut woods (CCHW) are the most suitable to enhance the effectiveness of ECs, since their marginal effects on ECWD imply that a 1% increase is correlated to 7.2- and 6.9-meter decrease in average CWD, respectively. Less relevant positive effects on the eligibility of a patch to be included in an EC are exhibited by grasslands (GRASS) and artificial woods (ARWO), whose marginal effects on ECWD are 5.81 and 5.33 m. The impacts of the covariates associated to scrub vegetation in dry areas and wetlands (SCRB),

and specialized herbaceous crops, agricultural and forest areas and uncultivated areas (HAFU), are definitely less important since their coefficients entail that an average increase of 1% is correlated to a 2.77- and 3.16- meter decrease in ECWD, respectively.

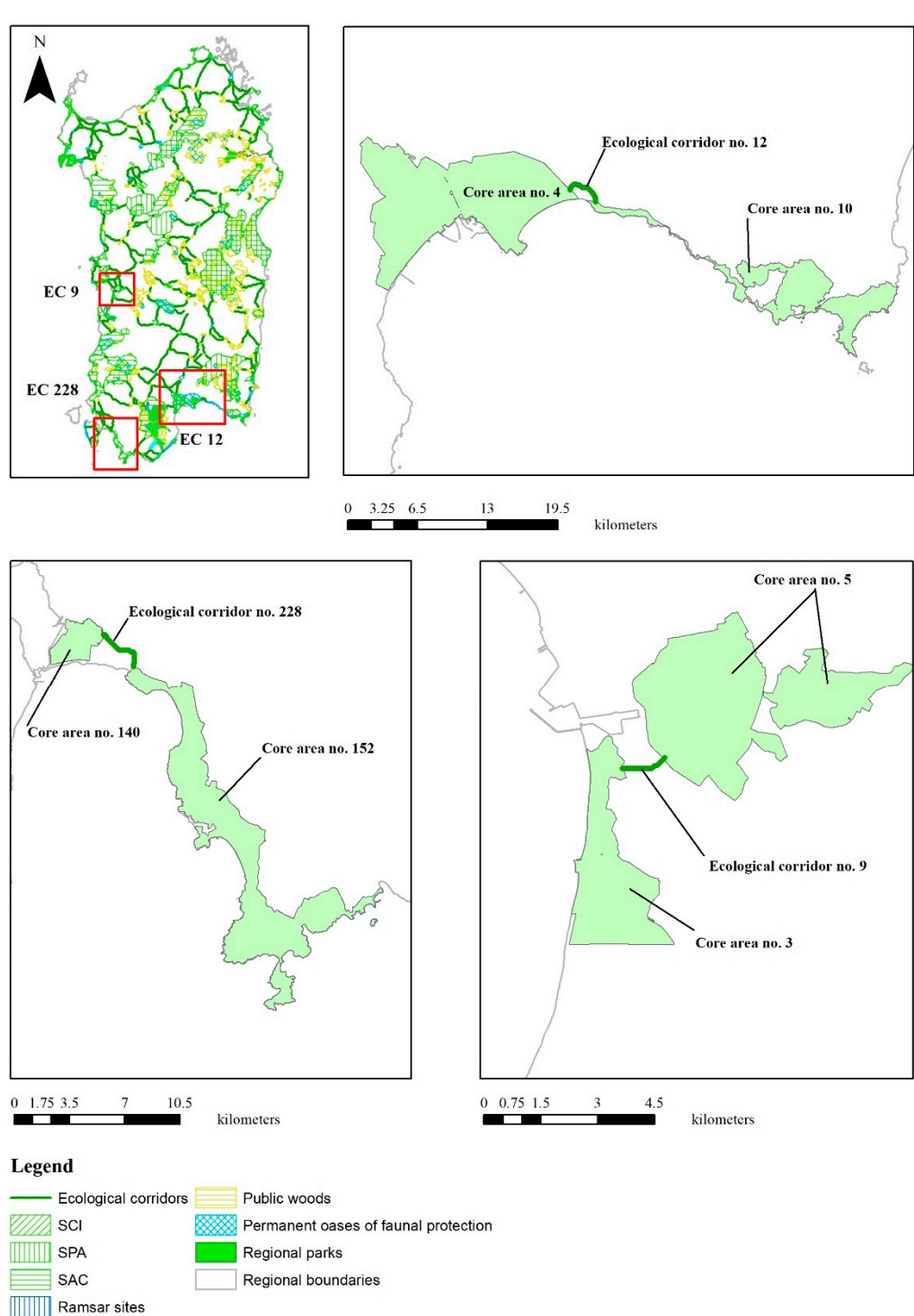

**Figure 10.** Spatial identification of the ECs which show the highest values of the CWD/ED index.

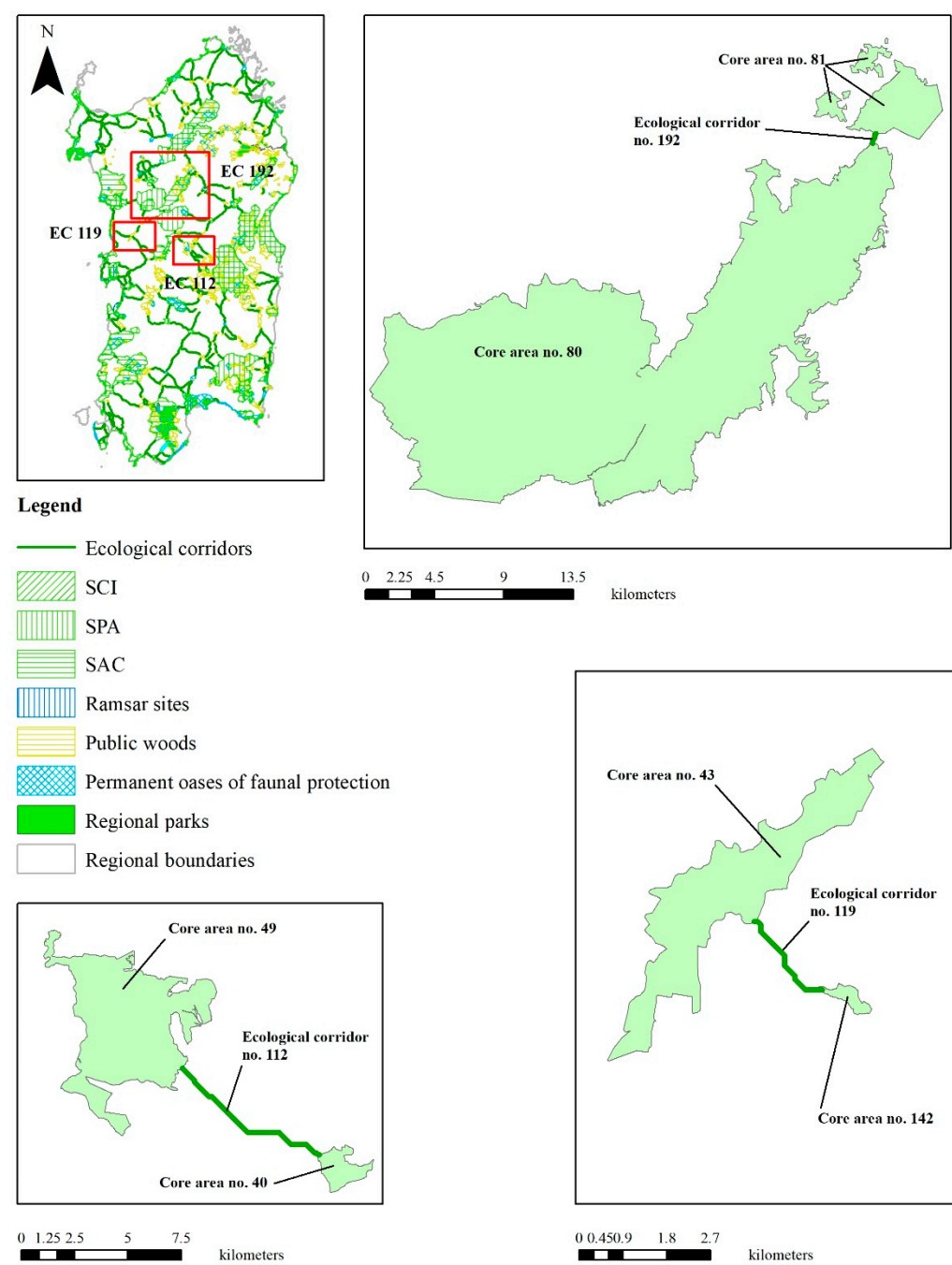

The codes of the core areas are identified in tab. 1

**Figure 11.** Spatial identification of the ECs which show the lowest values of the CWD/LCP index.

These outcomes make it easy to identify relevant planning policy implications related to the enhancement of the regional network of protected areas through the protection and the improvement of its ECs.

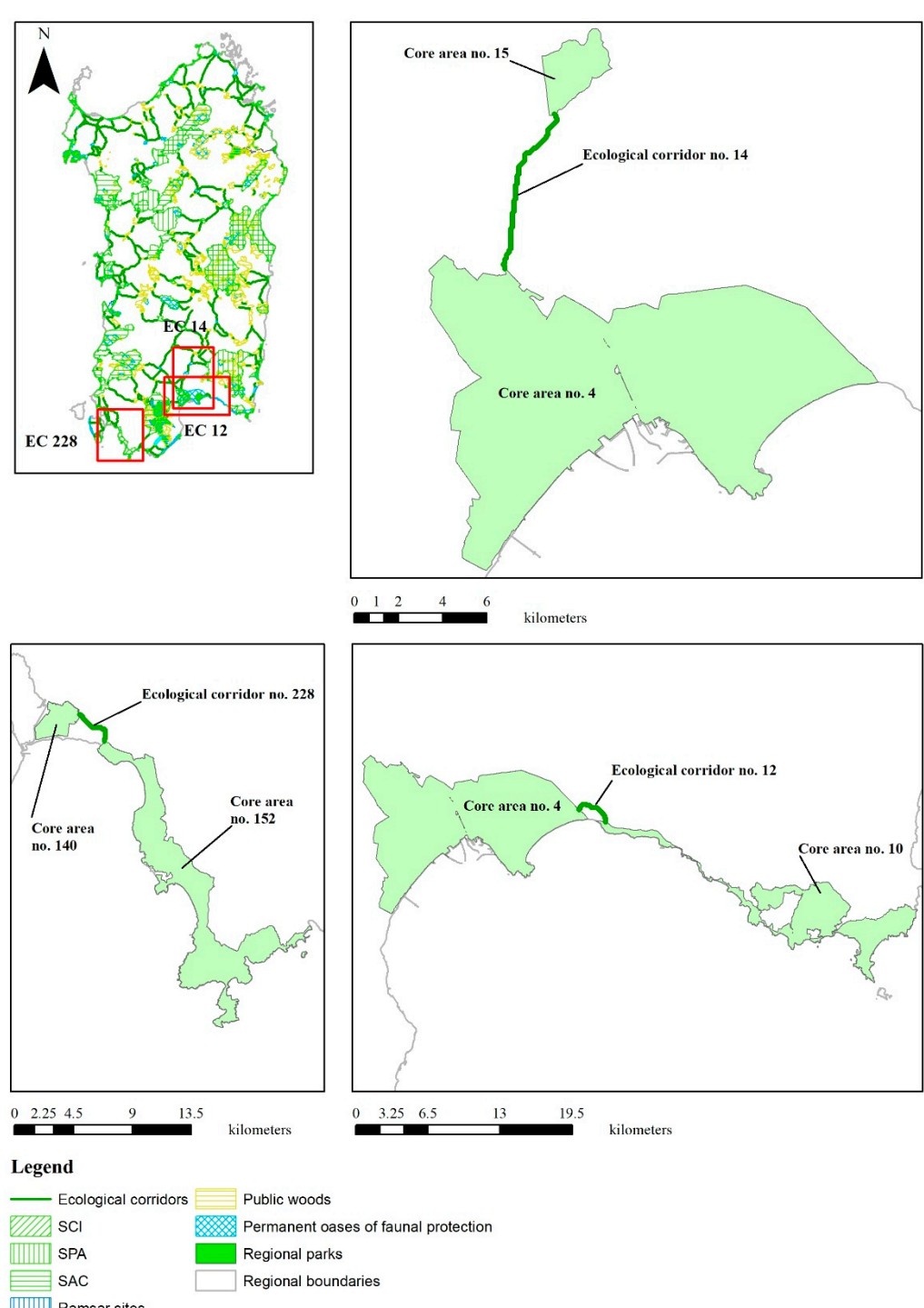

The codes of the core areas are identified in tab. 1

**Figure 12.** Spatial identification of the ECs which show the highest values of the CWD/LCP index.

**Table 3.** Definition of variables and descriptive statistics related to model (3).

| Variable | Definition | Mean | St.dev. |
|---|---|---|---|
| ECWD | Cost–weighted distance of a patch included in an EC (km) | 4947.08 | 2865.14 |
| SCRB | Scrub vegetation in dry areas and wetlands in a patch included in an EC (ha) | 16,962.44 | 26,775.40 |
| WOOD | Woodlands in a patch included in an EC (ha) | 18,038.76 | 29,513.47 |
| GRAS | Grasslands in a patch included in an EC (ha) | 18,879.67 | 27,314.39 |
| CCHW | Cork and chestnut woods in a patch included in an EC (ha) | 3190.58 | 12,865.24 |
| SPTC | Specialized and tree crops in a patch included in an EC (ha) | 3107.70 | 11,326.36 |
| ARWO | Artificial woods in a patch included in an EC (ha) | 4721.13 | 16,500.74 |
| HAFU | Specialized herbaceous crops, agricultural and forest areas, and uncultivated areas, in a patch included in an EC (ha) | 23,207.82 | 31,984.68 |
| ALTD | Control variable which represents the average altitude in a patch included in an EC (m) | 365.36 | 275.76 |

**Table 4.** Estimate of multiple linear regression model (3).

| Variable | Coefficient | *t*-Statistic | *p*-Value |
|---|---|---|---|
| SCRB | −2.77172 | −1.60534 | 0.108428 |
| WOOD | −7.20867 | −4.16805 | 0.000031 |
| GRAS | −5.80510 | −3.35834 | 0.000785 |
| CCHW | −6.91227 | −3.49271 | 0.000479 |
| SPTC | 7.70003 | 3.69729 | 0.000218 |
| ARWO | −5.33205 | −2.85482 | 0.004309 |
| HAFU | −3.16692 | −1.84476 | 0.065081 |
| ALTD | 1.45191 | 22.37541 | 0.000000 |
| Adjusted R-squared: 0.7123 | | | |

## 4. Discussion

The results of model (3), shown in the third section, make it possible to assess if, and to what extent, the current zoning code of the RLP can represent a solid basis for effectively protecting ECs, and highlight important implications for spatial planning practice.

The transition from agricultural to forest land uses, which should be supported by financial grants aimed at compensating differential yields, is associated with a decrease in CWD, and, as a consequence, a strengthening in the EC's spatial structure.

Planning measures focused on agroforestry transition are more straightforward and easier to implement as regards the areas classed as HAFU (as for specialized herbaceous crops, agricultural and forest areas, and uncultivated areas), and, even more, with reference to zones classed as SPTC (as for temporary crops related to other permanent crops). On the other hand, land cover transitions from intensive agricultural production areas, characterized by high crop yields, to less profitable woodlands (WOOD) or cork and chestnut woods (CCHW), can hardly be compensated by means of public grants, due to the size of the needed financial effort [56]. As regards intensive agricultural production zones, agroforestry transition should be implemented through a cooperative and integration-oriented policy by the involved public administrations at different spatial scales, in terms

of technical expertise and financial feasibility assessment [57–59]. For instance, in the case of goat and sheep farming, land cover change from grazing land to wooded areas can be effectively financed through public grants, so as to mitigate or even fully compensate for the yield decrease implied by such transition. Different is the case of cattle grazing areas, characterized by very high yields, whose transition would possibly generate relevantly, and even dramatic and destabilizing, impacts on the regional livestock economy, since wooded pasture, such as the Spanish dehesa, is not economically suitable for cattle farming.

Furthermore, afforestation intensity should be carefully assessed. As per Li et al. [60], an increasing trend in wooded areas may possibly be associated with a progressive rise in the ratio of costs to benefits of afforestation processes. Feng et al. [61] show that the unbalanced development of woodlands is likely to put at risk food safety. This implies that trade-offs between the provision of different ecosystem services and their economic and social benefits should be analyzed in detail.

A specific assessment of the question of afforestation, based on the land cover transition from farming to forestry, is proposed in a study related to social and economic factors driving from croplands to afforestation, which are particularly focused on the identification of the determinants concerning policy-making decisions [62]. From this standpoint, the perception of benefits coming from farming is an important obstacle regarding afforestation [63]. This is basically due to the farmers' strong perception of the positive effects related to the non-market value generated by flexible farming-related practices, and to their unwillingness to lose their durable expertise, which in their view is likely to be more important than the expected increase in income coming from afforestation [62]. The transition from intensive farming to forest land cover differs significantly from afforestation which originates from extensive croplands [64]. In the former case, a transition is quite unlikely, whereas it is much more probable in the latter since the expected income from forestry is likely to exceed the income coming from extensive farming, while intensive farming, which develops from permanent agriculture through high-yielding crops, is expected to have the highest rent [65]. Extensive and intensive farming should be targeted in terms of planning measures to decrease LST, based on incentive schemes. Agricultural farmers may possibly engage in afforestation, and, by doing so, disengage from low-income farming. The incentive effectiveness is likely to be identified in afforestation coming from transitions from mosaic farmlands and grazing lands, whereas it is quite unlikely that this is the case regarding intensive agriculture [56]. Moreover, the expansion of forest areas throughout rural zones featured by high-income farming should be carefully assessed by planning agencies in terms of financial feasibility as much as they should carefully consider the negative impact of afforestation on the traditional rural framework in terms of economic, social and landscape degradation [66].

All in all, planning policies and measures to strengthen the operational capacity and effectiveness of the regional network of protected areas through the protection and the improvement of the spatial framework of its ECs have to be studied, structured and implemented by focusing on the ruling concept that habitat quality, ecological integrity, and ecosystem conditions have to be enhanced and boosted-up [67].

Since the nodes of the networking spatial structure of the regional GI are identified with the system of the regional protected areas, whose protection regime implies conservation and enhancement of habitat quality, ecological integrity and ecosystem services, strengthening such spatial structure entails the establishment and implementation of planning policies aimed at extending to ECs the protection regime related to protected areas. Indeed, the locations of ECs are generally characterized by less restrictive planning rules than protected areas, in terms of conservation of nature and ecosystems. For example, as for the sites of the Natura 2000 network, which represents a relevant share of the nodes of the regional GI, conservation measures are established and implemented, under the provisions of the Habitats and Birds Directives, with reference to the nodes of the network, that is SCIs, SACs and SPAs, whereas the edges, that is ECs, are exposed to hazards coming from residential settlements and industrial activities. Urbanization and land-taking processes

should be targeted by planning policies aimed at protecting ECs as fundamental elements of the regional GI. Mitigation or prevention of land uptake and soil sealing should develop from interdisciplinary scientific bases, and should provide the public administrations with analytical technical skills in order to define policy measures aimed at implementing regional and local development processes based on the improvement of habitat quality, ESs provision and ecological integrity. The extension of the protection regime of the Natura 2000 Network as a point of reference to define and implement a comprehensive planning approach based on the conservation of habitats and species can be effectively supported by the increase in the number of the established Natura 2000 sites, which can be the outcome of thorough lobbying pressures by the local municipalities on the regional, national and European public authorities, based on sound scientific motivation, analysis and assessment of the connection between land-taking and soil-sealing processes, and qualitative and quantitative decrease in ESs provision [29]. From this standpoint, afforestation and reforestation can be considered highly supportive and complimentary planning policies, as discussed above.

Planning policies aimed at strengthening the ECs spatial structure should take account of the possible trade-offs between the supply of different ESs types. As for the Millennium Ecosystem Assessment [68], habitat quality, biodiversity flows opportunities and ecological integrity are classed as supporting and regulating ESs, whose supply can very possibly compete with provisioning and recreational ESs, such as cattle and farming production, and sport and leisure infrastructure. As discussed above, the transition from agricultural to forest land uses is a typical example of how to address a trade-off issue concerning provisioning and supporting ESs by increasing the supply of supporting ESs and, by doing so, strengthening the spatial structure of ECs., Kovács et al. address this question by assessing ESs trade-offs as regards three protected areas located in Hungary [69].

## 5. Conclusions

Building on a methodological approach defined by Cannas [27–30], in this study the issue of the identification of ECs is discussed, and ECs are detected with reference to the regional spatial context of Sardinia. Moreover, public policy-makers are provided with sound effective support for the conservation and enhancement of regional networks of NPSs and ECs on the basis of the implications of the methodology implementation.

Such methodology, which entails the mixed use of the least-cost path and regression models, shows two innovative aspects. First, the data used in this study are open-source and easily accessible to decision-makers, planners and research scholars. Therefore, the methodological approach is cheap in terms of cost and time. Secondly, the current literature mainly focuses on how to identify ECs within either regional or national contexts, whereas the normative aspects are often understated. Indeed, assessing if, and to what extent, spatial zoning codes can be used as a basis to implement regulations aimed at protecting ECs is still an under-explored research theme.

From this standpoint, it has to be highlighted that the methodological approach defined and implemented in this study can be easily exported to the EU local, regional and national scales, since spatial databases consistent with each other are available for Sardinia and Italy as for the other regions and countries. Moreover, such methodology shows a flexible structure that makes it easy to adjust in real-time the ever-evolving process of identification, protection and enhancement of the spatial structure of ECs. This is particularly relevant as regards ESs provision and related data, expertise involvement and information retrieval.

Habitat-suitability and ecological-integrity maps are based on scientists' and practitioners' expertise. Indeed, such expertise provides the public administrations with sound and effective advice on habitat-suitability species-specific values, which works as a foundation for the elaboration of habitat-suitability maps, and on ecological integrity values, which is the basis for the definition of ecological-integrity maps. Nevertheless, the use of these maps suffers from a certain degree of subjectivity, which implies controversial results in their application, especially if scientific and technical knowledge is lacking.

This is an important issue as regards future research on the methodology defined in this study, which can be identified with reference to the ecological integrity map related to the Sardinian regional context. As was described in Section 2.2, such ecological integrity map is based on experts' judgments concerning the supply potential of CLC land cover classes with reference to seven ESs, that is, abiotic heterogeneity, biodiversity, biotic waterflows, metabolic efficiency, energy capture, reduction of nutrient loss and storage capacity, identified by Burkhard et al. [43]. At present, a systemic and detailed scientific and technical information concerning the relations between CLS classes and ESs provision is not available for Sardinia and for the other Italian regions, and, that being so, the methodological approach defined in this study, whose prototypical implementation is based on experts' judgments reported in the Burkhard et al.'s article, will be usable in the real world if, and only if, scientific and technical knowledge on the supply potential of CLC classes in terms of ESs provision will be readily and transparently available in open -data format. This implies an important further effort in theoretical and applied research on ESs and land covers and land uses, and in the implementation of these outcomes in the planning practices of the bodies at the different levels of the public administration.

**Author Contributions:** F.I., F.L. and C.Z. collaboratively designed this study and jointly took care of Sections 1, 2.1 and 4. F.I. wrote Section 2.2. C.Z. wrote Sections 2.3 and 3.2. F.L. wrote Section 3.1. F.I. and F.L. jointly wrote Section 5. All authors have read and agreed to the published version of the manuscript.

**Funding:** The study was funded by the Research Program "Paesaggi rurali della Sardegna: pianificazione di infrastrutture verdi e blu e di reti territoriali complesse" [Rural landscapes of Sardinia: Planning policies for green and blue infrastructure and spatial complex networks], funded by the Autonomous Region of Sardinia for the period 2019–2021, under the provisions of the call for the presentation of "Projects related to fundamental or basic research" of the year 2017, implemented at the Department of Civil and Environmental Engineering and Architecture (DICAAR) of the University of Cagliari, Italy.

**Conflicts of Interest:** The authors declare no conflict of interest.

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
