# Peer review of "Mapping of Ecological Corridors as Connections between Protected Areas: A Study Concerning Sardinia, Italy"

_sustainability, doi:10.3390/su14116588_

Round 1
Reviewer 1 Report
The authors examined the mapping of ecological corridors as connections between protected areas in Italy with addressing methodological approach. There are some amendments explained below:
- The introduction can be re-organized by the background of the topic with connected their previous published studies, and more clearence on the aim of the study. Also, the authors indicated there are 4 section in the methodology, but section 1 is little bir missing. Threfore, re-organized and clear the aim and construction of the study.
- Construction flow of the study can be illustrated by the authors giving with the references (like fig2)
- Figure 1 needs more resolution or more contrast or change the colors to clear identification of the fields.
- In materials methods, some introduction were used, please moved them into introduction.
- what is tha advantages and drawbacks this concept, and improvable aspects of this structure.
- some examples on the planning policies and measures to strengthen operational capacity of the protected areas can be given.
Author Response
The authors examined the mapping of ecological corridors as connections between protected areas in Italy with addressing methodological approach. There are some amendments explained below:
- The introduction can be re-organized by the background of the topic with connected their previous published studies, and more clearence on the aim of the study. Also, the authors indicated there are 4 section in the methodology, but section 1 is little bir missing. Threfore, re-organized and clear the aim and construction of the study.
As recommended by the Reviewer, the Introduction was reorganized, the background of the topic of the submitted manuscript discussed and the objective of the study stated in a clear-cut way. The reorganization of the Introduction is in line with point 3 raised by Reviewer #2 (lines 42-147).
- Construction flow of the study can be illustrated by the authors giving with the references (like fig2)
Following the Reviewer’s suggestion, the scheme of the methodological approach implemented in this study is represented in Figure 1 of Section 2. “Materials and methods”.
- Figure 1 needs more resolution or more contrast or change the colors to clear identification of the fields.
Done.
- materials methods, some introduction were used, please move them into introduction.
Following the Reviewer’s suggestion, a part of Subsection 2.1 “Study area” of Section 2. “Materials and methods” of the previous version of the study was moved to the Introduction (lines 94-127).
- what is the advantages and drawbacks this concept, and improvable aspects of this structure.
According to the Reviewer’s recommendation, in a newly-added “Conclusions” Section future research directions are described, in the light of the study’s strengths and drawbacks (lines 708-773).
- some examples on the planning policies and measures to strengthen operational capacity of the protected areas can be given.
Following the Reviewer’s suggestion, examples of planning policies and measures were added to the Discussion Section (lines xxx-xxx). Incidentally, the previous “Discussion and conclusions” Section was split into two sections by separating “Discussion” and “Conclusions”, according to the Reviewer 2’s recommendation.
Reviewer 2 Report
This work tries to provide an exploratory analysis of the ecological corridors as green infrastructures to protect wildlife habitat. The topic is interesting. However, in my opinion the paper has shortcoming in the text regards to structure’s on the paper. And I think that the manuscript seems that the contents and intentions of the study are no sufficiently explained. So, It is potentially acceptable, but there is room for improvement.
- Abstract: Please don’t use acronyms here. I recommend the authors to clarify the purpose of this study. The methods also need to be described in more detail in here. Please summarize the main results of this study based on the research questions. You don’t need to describe very general information, such asm," An important set of Ecosystem services (ESs) provided by Green infrastructures (GIs) con-8 sists of habitats and species protection and improvement, which coincides with biodiversity conser-9 vation and enhancement.”
- Keywords: They are very redundant with your title and redundant among them; try to add some others, for example, keywords related to methodology.
- Introduction: Please try to explain from the beginning what is your paper about and which is your main contribution. I think you don’t need to describe the methodology in the introduction section, such as “ Building on methodological approach~~~~”. I think you just briefly the research flow using the explanation about the structured four section, but I think it is better to describe What is main purpose of this study, and objectives of this study using the research questions. You should clarify the purpose and objectives of the study. Research flow including the explanation about the structured section should be described in the methodology section.
- Methodology: Please revised this section according to the 4 sections which the authors mentioned in the Introduction section. I think, it will be easier to understand. And please describe the data selections with table.
- Results: Of course, you should be described the results according to the 4 sections which you mentioned before. If that, it looks better organized. How did you adopt the weight or resistance values, regarding to different land use, to connect ecological corridors between the core areas. Among the core aares, which one is source area or destination area? And how about R2 values from the regression model? I am not sure the statistical significance from these results.
- Discussion: There needs to be a separate discussion about the noble of this study, limitation, and future research direction from the methodological perspective.
Author Response
This work tries to provide an exploratory analysis of the ecological corridors as green infrastructures to protect wildlife habitat. The topic is interesting. However, in my opinion the paper has shortcoming in the text regards to structure’s on the paper. And I think that the manuscript seems that the contents and intentions of the study are no sufficiently explained. So, It is potentially acceptable, but there is room for improvement.
- Abstract: Please don’t use acronyms here. I recommend the authors to clarify the purpose of this study. The methods also need to be described in more detail in here. Please summarize the main results of this study based on the research questions. You don’t need to describe very general information, such asm," An important set of Ecosystem services (ESs) provided by Green infrastructures (GIs) con-8 sists of habitats and species protection and improvement, which coincides with biodiversity conser-9 vation and enhancement.”
Following the Reviewer’s suggestion, the abstract was reorganized. In particular, the objectives of the study and the methodologies description were added (lines 14-37).
- Keywords: They are very redundant with your title and redundant among them; try to add some others, for example, keywords related to methodology.
As recommended by the Reviewer, some keywords were changed and keywords related to the methodology were added (lines 38-39).
- Introduction: Please try to explain from the beginning what is your paper about and which is your main contribution. I think you don’t need to describe the methodology in the introduction section, such as “ Building on methodological approach~~~~”. I think you just briefly the research flow using the explanation about the structured four section, but I think it is better to describe What is main purpose of this study, and objectives of this study using the research questions. You should clarify the purpose and objectives of the study. Research flow including the explanation about the structured section should be described in the methodology section.
As recommended by the Reviewer, the Introduction was reorganized, the background of the topic of the submitted manuscript discussed and the objective of the study stated in a clear-cut way. The reorganization of the Introduction is in line with point 3 raised by Reviewer #1 (lines 42-147).
- Methodology: Please revised this section according to the 4 sections which the authors mentioned in the Introduction section. I think, it will be easier to understand. And please describe the data selections with table.
As suggested by the Reviewer, a table reporting the data selection was added.
- Results: Of course, you should be described the results according to the 4 sections which you mentioned before. If that, it looks better organized. How did you adopt the weight or resistance values, regarding to different land use, to connect ecological corridors between the core areas. Among the core aares, which one is source area or destination area? And how about R2 values from the regression model? I am not sure the statistical significance from these results.
Following the Reviewer’s suggestion, references to the goodness of fit with respect to the value of the correlation coefficient of the regression estimate (adjusted R-squared) were added to the Results Section and the value of the correlation coefficient was displayed in Table 3. Moreover, the Results Section was reorganized according to the four phases mentioned in Section 2.2 “Spatial identification of the ecological corridors” (lines 466-481).
- Discussion: There needs to be a separate discussion about the noble of this study, limitation, and future research direction from the methodological perspective.
Following the Reviewer’s suggestion, the previous “Discussion and conclusions” Section was split into two sections by separating “Discussion” and “Conclusions”, according to the Reviewer’s recommendation. In the conclusions future research directions are described, in the light of the study’s strengths and drawbacks (lines 708-773).

Reviewer 3 Report
- The citation style of the references needs to be carefully checked. For example, in section 2.1, “2.1 Study area [Error! Reference source not found.]”. And reference 42.
- Personally, I do not recommend citing references in the Abstract.
-
Make sure all abbreviations are written out in full the first time used. For example, RLP in Line 76, Page 2.
Author Response
- The citation style of the references needs to be carefully checked. For example, in section 2.1, “2.1 Study area [Error! Reference source not found.]”. And reference 42.
As recommended by the Reviewer, the references were verified in the text.
- Personally, I do not recommend citing references in the Abstract.
As recommended by the Reviewer, citations were removed.
- Make sure all abbreviations are written out in full the first time used. For example, RLP in Line 76, Page 2.
As recommended by the Reviewer, the abbreviations were verified in the text.
